# Transformer-Based Large Language Models Are Not General Learners: A Universal Circuit Perspective

## Abstract

Large Language Models (LLMs) have demonstrated remarkable proficiency across diverse tasks, evoking perceptions of "sparks of Artificial General Intelligence (AGI)" (Bubeck et al., 2023). A key question naturally arises: *Can foundation models lead to AGI?* In this work, we try to answer this question partially by formally considering the capabilities of Transformer-based LLMs (T-LLMs) from the perspective of universal circuits. By investigating the expressive power of realistic T-LLMs as universal circuits, we show that a T-LLM of size $\mathrm{poly}(n)$ cannot perform all the basic operators of input length $O\left(\mathrm{poly}(\log n)\right)$. We also demonstrate that a constant-depth-$\mathrm{poly}(n)$-size log-precision T-LLM cannot faithfully execute prompts of complexity $n$. Our analysis provides a concrete theoretical foundation that T-LLMs can only be universal circuits for limited function classes, or in other words, T-LLMs are not general learners. Furthermore, we exhibit that a constant-depth-$\mathrm{poly}(n)$-size log-precision T-LLM can memorize $O\left(\mathrm{poly}(n)\right)$ instances, which could partially explain the seeming inconsistency between LLMs' empirical successes and our negative results. To the best of our knowledge, our work takes the first step towards analyzing the limitations of T-LLMs as general learners within a rigorous theoretical framework. Our results promote the understanding of LLMs' capabilities and highlight the need for innovative architecture designs beyond Transformers to break current limitations.

## 1 Introduction

Large Language Models (LLMs) (Touvron et al., 2023; OpenAI, 2023a;b) have achieved unprecedented success across a wide range of tasks such as code generation (Chen et al., 2021; Zhang et al., 2022), mathematical problem solving (Frieder et al., 2023), reasoning (Webb et al., 2023; Hao et al., 2023), knowledge retrieval (Sun et al., 2023; Zhao et al., 2023) etc. This remarkable success triggers a notable shift in the research priorities of the artificial intelligence community. These impressive empirical achievements fuel an expectation that LLMs are "sparks of Artificial General Intelligence (AGI)". However, some evaluation results have also presented confusing instances of LLM failures, including some in seemingly trivial tasks. For example, GPT-4 is able to solve some mathematical problems in IMO that could be challenging for graduate students, while it could make errors on arithmetic problems at an elementary school level in some cases (Bubeck et al., 2023). This intriguing inconsistency raises doubts on LLMs' capabilities for general problem solving, an essential feature of AGI, and leads to a fundamental question: *Can LLMs be general learners?*

There have been extensive works that attempt to probe the boundary of the LLMs, showing the limitations of LLMs on abstract reasoning (Gendron et al., 2023), counterfactual simulatability (Chen et al., 2023), reverse direction generalization (Berglund et al., 2023), etc. While these empirical results provide evidence that the LLMs may not be general learners, they could not give a clear boundary of the LLM capabilities and thus a definitive answer to the core question. To have a rigorous analysis for the general learner question, an inevitable concern is whether the underlying models of LLMs, i.e., Transformers, have the power to represent general learners in theory. While ideal Transformers are Turing complete (Pérez et al., 2019; 2021) and can thus handle all computable problems, realistic Transformers have much limited capacities. It is unknown whether the realistic Transformers are expressive enough to be general learners (Hahn, 2020; Hao et al., 2022;

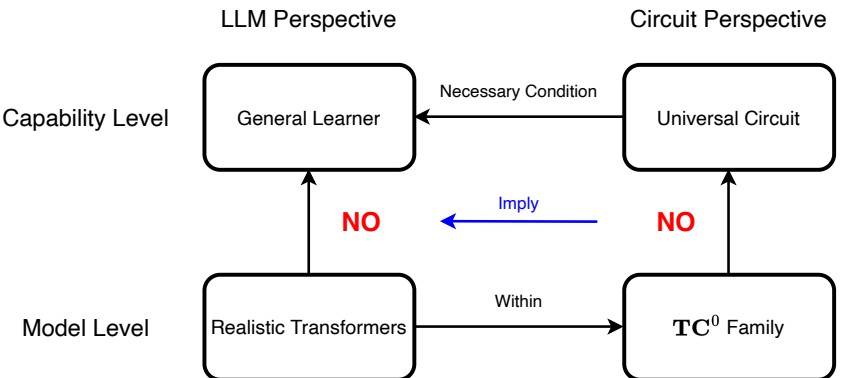

Figure 1: Our paradigm for analyzing whether T-LLMs can be general learners from the universal circuit perspective. In the capacity level, our goal is to analyze whether a model can be a general learner from the perspective of LLMs. A necessary condition is that the model is as powerful as a universal circuit for the set of all polynomial-size circuits. In the model level, we consider the realistic Transformers, which are within $\mathbf{TC}^0$ family from the circuit perspective. By circuit complexity theory, no circuits $\mathbf{TC}^0$ family can implement universal circuits for all polynomial-size circuits. This implies that no models based on realistic Transformers can be general learners.

Merrill et al., 2022; Merrill & Sabharwal, 2023). The key point is to identify a clear boundary of how powerful a model should be as a general learner.

This work takes the first step towards rigorously answering the fundamental question by considering a theoretical boundary of model capacity to be general learners. We first build a formal framework of universal circuits, under which we can conduct a rigorous analysis on the limitations of LLM capabilities. A universal circuit takes an encoding that describes a circuit and an input to the circuit, simulating the computation of the encoded circuit on the input. Ideally, an LLM takes a prompt and a task input, predicting the task solution given the input by following the inference steps informed by the prompt. Therefore, both universal circuits and LLMs can be seen as a form of universal computation. Furthermore, we can build correspondences between the concepts of LLMs and universal circuits, e.g., prompt and circuit encoding, task input and circuit input, single inference step and gate, etc. These correspondences allow us to characterize the capabilities of LLMs by the power of their corresponding universal circuits. In this work, we take this approach to investigate whether LLMs are general learners. To clarify further, for an LLM being general learner, an essential capability to understand and execute all the prompts within some tractable complexity. In the language of universal circuits, given any input string and any description of circuits of polynomial size, the universal circuit corresponding to a general learner should be able to simulate the computation of the input circuit given the input string. In other words, if an LLM is a general learner, it is at least as powerful as a universal circuit for the class of polynomial-size circuits. Under this reduction, we derive a necessary condition that LLMs can be general learners: the class of circuits corresponding to LLMs contains a universal circuit for the class of all polynomial-size circuits.

We then consider whether LLMs of the current paradigm, i.e., Transformer-based LLMs (T-LLMs) can be general learners. We focus on a realistic setting of T-LLMs, i.e., T-LLMs based on constant-depth-polynomial-size log-precision Transformers (Merrill et al., 2022; Merrill & Sabharwal, 2023). This setting captures the essential restrictions on a T-LLM in practice, i.e., finite precision, limited depth, and limited model size. This setting has been followed by a line of works on theoretical limitations of realistic Transformers to depict which complexity classes they belong to (Hahn, 2020; Hao et al., 2022; Merrill et al., 2022; Merrill & Sabharwal, 2023). We present two impossibility results: a constant-depth-$\mathrm{poly}(n)$-size log-precision T-LLM cannot perform all the basic operators of input length $O\left(\mathrm{poly}(\log n)\right)$ and it cannot faithfully execute prompts of complexity $n$. The two impossibility results are derived via our reduction from the capabilities of LLMs to the existence of universal circuits. More specifically, we prove the two impossibility results by showing the non-existence of universal circuits within the realistic T-LLM class for all the Boolean functions of input length $O\left(\mathrm{poly}(\log n)\right)$ and all the circuits of polynomial size, respectively. In plain words, these

results imply that realistic T-LLMs have an inherent limitation to be general learners. Figure 1 summarizes our paradigm to analyze whether T-LLMs can be general learners from the universal circuit perspective. We also show that the negative results do not contradict the observed empirical success of LLMs. We demonstrate the possibility that LLMs could partially solve complex tasks by memorizing some instances when the tasks could not be solved in general cases.

We summarize the contributions of this work from three aspects. First, this work makes an attempt to theoretically consider the fundamental problem, i.e., whether LLMs can be general learners. We derive a negative answer that realistic T-LLMs cannot be general learners. To the best of our knowledge, we are the first to draw the conclusion in a rigorous manner. Second, this work stresses the necessity of developing novel architectures in order to achieve powerful models. This highlights architecture design as an important future direction. Additionally, this work develops a comprehensive framework of universal circuits to model the capabilities of LLMs. Besides the analysis on general learners in this work, this framework could be employed to formally analyze a wide variety of capabilities of LLMs.

The remaining sections of the paper are organized as follows. Section 2 first introduces the essential concepts on universal circuits and then describes our framework for analyzing LLM capabilities from the perspective of universal circuits. Section 3 shows that T-LLMs cannot be general mappings and have very limited capabilities to perform basic operators. Section 4 shows that T-LLMs cannot be general learners. The results of Sections 3 and 4 give a negative answer to the fundamental question. Section 5 shows the possibility that T-LLMs can memorize instances. This provides a possible explanation for why T-LLMs that are not general learners can achieve empirical success. Section 6 discusses some related works. Section 7 summarizes the paper, points out the limitations of our results, and highlights some interesting directions for future research.

**Notation.** We denote the alphabet by $\Sigma$. We assume $\Sigma = \{0, 1\}$ throughout the remainder of the paper unless otherwise stated. We denote the set of all circuits having inputs of length $n$ and outputs of length $m$, of size bounded by $s$ and depth bounded $d$, and consisting of gates from $\mathcal{F}$ by $\Gamma(n, m, s, d, \mathcal{F})$. Specially, $\Gamma(n, m, s, \infty, \mathcal{F})$ represents all circuits having $n$ inputs and $m$ outputs, of size bounded by $s$ and depth unbounded. We denote the set of all Boolean functions of input length $n$ by $B_n$, i.e., $B_n = \{f_n \mid f_n : \{0, 1\}^n \mapsto \{0, 1\}\}$.

## 2 A UNIVERSAL CIRCUIT PERSPECTIVE ON LLMS

A general learner has the essential ability to solve *all* tractable problems, i.e., problems that can be solved efficiently with some proper algorithms. Considering humans as general learners, we anticipate that individuals have the ability to address manageable problems, which could be solved in a reasonable number of steps, by following specific instructions. For LLMs, the essential ability as general learners corresponds to the capability to solve *all* problems for which there exist prompts leading to efficient inference processes. In order to rigorously examine if LLMs have the essential capability as general learners, we require a sufficiently expressive framework capable of describing the property in formal language. We also want the framework to capture the features of LLMs' inference processes, providing a proper abstract model for LLMs and building connections between the capabilities of LLMs and the capacities of their underlying models (for the LLMs of the current paradigm, the underlying models are Transformers).

To address the requisite for a sufficiently expressive framework that accurately encapsulates the capabilities of LLMs, we turn to the concept of universal circuits. We show in Section 2.1 that universal circuits are sufficiently expressive to cover complexity classes for realistic general learners. We also illustrate in Section 2.2 that universal circuits capture essential features of LLMs' inference, serving as good alternatives when discussing the capabilities of LLMs as well as connecting the capabilities of LLMs and the capacities of Transformers. To further elaborate, we begin with a concise introduction to universal circuits.

### 2.1 UNIVERSAL CIRCUIT

Universal circuits (Valiant, 1976; Cook & Hoover, 1985; Alhassan et al., 2020) are general-purpose circuits that can simulate all circuits in some set. More concretely, a universal circuit takes an input consisting of two parts : one part encodes a target circuit to be simulated and the other is the input

to the target circuit. The circuit encoding is an input string that describes everything essential of the target circuit following some encoding rules. Informally, we can think that the universal circuit is configured according to the encoding and the configured universal circuit can simulate the target circuit on all inputs. Definition 1 is a formal description of the universal circuit.

**Definition 1** (Universal Circuit). *A circuit $\alpha : \Sigma^n \times \Sigma^l \mapsto \Sigma^m$ is called a universal circuit for a set of functions $\mathcal{F} \subseteq \{f \mid f : \Sigma^n \mapsto \Sigma^m\}$ for all $f \in \mathcal{F}$ if and only if for each $f \in \mathcal{F}$, there exists a encoding $e_f$ such that $\alpha(x, e_f) = f(x)$ for all $x \in \Sigma^n$.*

We emphasize that the universality of a universal circuit always depends on a specific circuit set in our discussion. Previous works on universal circuits typically focus on the universality for very general circuit sets, such as the set of all circuits with input length $n$ and size $s$ and the set of all circuits with input length $n$, size $s$, and depth $d$. However, we can also consider circuits that are universal circuits for restricted circuit sets. An extreme example is that every circuit can be seen as a universal circuit for the set that contains a single circuit, i.e., the circuit itself, with an empty string as the circuit encoding. From this viewpoint, we can shift our focus from assessing the general capabilities of a model to analyzing the extent of generality in the circuit set that the model, when viewed as a universal circuit, can simulate. For instance, consider whether LLMs have the capability to perform both deductive reasoning and inductive reasoning. Suppose that there are two circuits that can perform deductive reasoning and inductive reasoning, respectively. Then we can consider whether the model class includes a model that simulates a universal circuit for a circuit set including the deductive reasoning circuit and the inductive reasoning circuit.

One of the most significant findings in the realm of universal circuits is Valiant's results on the existence of universal circuits: there exists a universal circuit of size $O(s \log s)$ for all the circuits of input length $n$ and size $s$ ($s \geq n$), where the gates are AND, OR, and NOT (Valiant, 1976). This result asserts that for any circuit set of polynomial size, a universal circuit of polynomial size can be constructed. Consequently, the class of polynomial circuits containing universal circuits proves to be sufficiently expressive for addressing all tractable problems, encompassing complexity classes relevant to practical tasks.

## 2.2 LLM AS UNIVERSAL CIRCUIT

An LLM takes an input, with a prompt (we can view the case without prompts as taking a special "empty" prompt which asks the LLM to solve the problem directly), to output the answer to the target problem based on the input. The prompt, intuitively, serves as a set of instructions, whether explicit or implicit, guiding the LLM in problem-solving, provided in the form of descriptions, code, demonstrations, etc. In an ideal scenario, the LLM faithfully adheres to the instructions within the prompt to solve the problem. This entails the LLM executing a pipeline informed by the prompt in a faithful manner (Lanham et al., 2023; Radhakrishnan et al., 2023). This suggests that an ideal inference process of the LLM aligns with the concept of universal computation, which simulates a set of computational devices when provided with the appropriate instructions or programs.

To rigorously formalize the above insight, we propose considering an LLM as a universal circuit for some circuit set $\mathcal{F}$. An LLM can perform some basic operators on small inputs. We denote the set of basic operators by $\mathcal{F}_0 \subseteq \Sigma^{n_0} \mapsto \Sigma^{m_0}$, where $n_0$ and $m_0$ are some constants. The basic operators serve as gates for the circuits in $\mathcal{F}$. A prompt describes a computation process composed of the basic operators (explicitly or implicitly). In the language of universal circuits, the prompt encodes a circuit where the gates are the basic operators. An inference process of faithful prompt following corresponds to a computation process of the universal circuit that simulates a target circuit given the target circuit encoding and a problem input.

Under the universal circuit framework, we can formally discuss the problem of whether LLMs are general learners. If an LLM is a general learner, it is essential that it can understand and faithfully follow the prompts of tractable complexity. From the perspective of universal circuits, the model of the general learner is at least a universal circuit for the set of all circuits within a polynomial size. Then we have a necessary condition for LLMs to be general learners.

**Proposition 1** (Necessary Condition to Be General Learners). *Suppose that $\mathcal{F}_0$ is a set of basic operators and $\{\Gamma_n\}$ is a class of LLMs where $\Gamma_n$ is a set of models for inputs of length $n$. The LLMs of the class are general learners only if for any integer $n$ and polynomials $m(\cdot), s(\cdot)$, there exists a universal circuit $\alpha_n \in \Gamma_n$ for the circuit set $\Gamma(n, m(n), s(n), \infty, \mathcal{F}_0)$.*

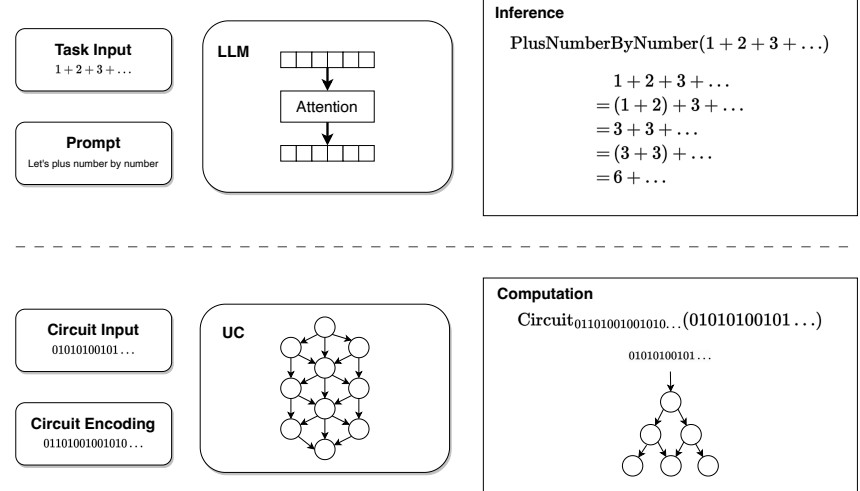

Figure 2: Correspondences between LLMs and Universal Circuits (UC). A task input in LLM corresponds to a circuit input in the universal circuit. A prompt in LLM corresponds to a circuit encoding in the universal circuit. An inference step in LLM corresponds to a gate in the simulated circuit. An inference process that executes the prompt given the task input in LLM corresponds to a computation process that simulates the computation of the encoded circuit given the circuit input. The inference structure corresponds to the underlying directed acyclic graph of the simulated circuit. Note that all the strings, diagrams, and graphs are only for the purpose of illustrating the concepts and do not reflect the exact models.

In the following sections, we focus on whether the current paradigm of LLMs, i.e., T-LLMs can satisfy the necessary condition of general learners. Unfortunately, under a realistic setting of T-LLMs, i.e., T-LLMs with log precision, constant depths, and polynomial sizes, we show T-LLMs do not satisfy this condition and thus cannot be general learners.

*Remark* 1. The universal circuit perspective is not to model the exact underlying computation process of LLMs. It is to offer a proper alternative that captures some significant features of LLMs such that we can analyze the capabilities of the extremely complex systems. In fact, building an abstract model to understand a system and explain related phenomena is a common practice in the research of very complex systems such as consciousness (Baars, 1993; Blum & Blum, 2022) and LLMs (Xie et al., 2021; Han et al., 2023). See Section 6 for a more detailed discussion on these related works.

## 3 LIMITATIONS OF T-LLMS AS GENERAL MAPPINGS

We first outline a loose boundary of T-LLMs' limitations as general mappings. The term "general mapping" of input length $m$ refers to a mapping that simulates all the mappings from $\Sigma^m$ to $\Sigma$ given the corresponding mapping encodings. Our results show that T-LLMs have very limited capabilities as general mappings, not covering those of input length $\text{poly}(\log n)$. Since $\text{poly}(\log n) = o(n)$, this means that T-LLMs could not be general mappings even for input length sublinear in $n$. Theorem 1 shows that no polynomial-size circuits, even without depth restrictions, can simulate all mappings of polylogarithmic input length.

**Theorem 1.** *Suppose that $\mathcal{F}_0 = B_2$. For any polynomials $l(\cdot)$, $s_0(\cdot)$, there exists a polynomial $p(\cdot)$ and an integer $n$ such that no circuit in $\Gamma(p(\log n) + l(n), 1, s_0(n), \infty, \mathcal{F}_0)$ is a universal circuit for the circuit set $B_{p(\log n)}$.*

For T-LLMs, we follow the realistic setting in Merrill & Sabharwal (2023). We consider log-precision, constant-depth, and polynomial-size Transformers: for Transformers whose input is of length $n$, the values at all neurons are represented with $O(\log n)$ bits, the depth is constant, and the number of neurons is $O(\text{poly}(n))$. This setting captures the design and the hardware restriction of realistic T-LLMs and is common in the literature on the theoretical power of T-LLMs (Hahn, 2020; Hao et al., 2022; Merrill et al., 2022; Merrill & Sabharwal, 2023). Theorem 2 is a result

from Merrill & Sabharwal (2023) that T-LLMs are within $\mathbf{TC}^0$ circuit family, the class of constant-depth, unbounded-fan-in, polynomial-size Boolean circuits with NOT, AND, OR, and threshold gates. Theorem 2 portrays the model capacity of T-LLMs by circuit complexity. This result allows us to analyze whether the model capacity is enough for the model capability from the circuit perspective, facilitating the use of tools from circuit complexity theory. For a comprehensive background on circuit complexity theory, we refer the reader to Arora & Barak (2009) and Vollmer (1999).

**Theorem 2** (Merrill & Sabharwal (2023)). *Any log-precision, constant-depth, and polynomial-size Transformer can be simulated by a $\mathbf{TC}^0$ circuit family.*

Since a circuit in $\mathbf{TC}^0$ is of polynomial size, we have that the $\mathbf{TC}^0$ class cannot implement general mappings of polylogarithmic input length. Combining with Theorem 2 that realistic T-LLMs are within $\mathbf{TC}^0$, we can conclude that realistic T-LLMs are not general mappings of polylogarithmic input length, as stated in Corollary 1.

**Corollary 1.** *For any integer $d_0$ and polynomial $s_0(\cdot)$, there exists an integer $n$ and a polynomial $p(\cdot)$ such that no log-precision Transformer-based LLM of depth $d_0$ and size $s_0(n)$ can perform all mappings of input length $p(\log n)$.*

The limitations of T-LLMs as general mappings reflect the limitations of the basic operators that can be performed by T-LLMs. Corollary 1 suggests that no realistic T-LLMs can perform all the basic operators of polylogarithmic input length.

*Remark* 2. In Theorem 1 and other theorems of this paper, we only discuss the case of single output, i.e., $m = 1$ for a circuit set $\Gamma(n, m, s, d, \mathcal{F})$. We can simply generalize to the multiple output case by connecting $m$ circuits in $\Gamma(n, m, s, d, \mathcal{F})$ in parallel. This slight simplification does not affect the results we draw on the limitations of LLMs.

# 4 LIMITATIONS OF T-LLMS AS GENERAL LEARNERS

We are now ready to discuss the limitations of LLMs as general learners. As discussed in Section 2.2, we can view the inference process of an LLM as the computation of a universal circuit. If the LLM is a general learner, the corresponding universal circuit must be universal enough to simulate all the circuits in a very general circuit set. For realistic problems, we deem the set as one consisting of all polynomial-size circuits. Based on these considerations, we reduce the problem of whether a realistic LLM can be a general learner to the problem of whether the realistic model class is expressive enough to contain such universal circuits. As Proposition 1 summarizes, a necessary condition for a model class to be general learners is that it can implement universal circuits for the circuit sets of polynomial size.

Our result on the limitations of T-LLMs as general learners comes from Proposition 1 and Theorem 2. On the one hand, T-LLMs are within the $\mathbf{TC}^0$ complexity family; on the other hand, general learners require at least as hard as $\mathbf{P}/\operatorname{poly}$-complete. In the field of circuit theory, it is known that $\mathbf{TC}^0$ is a subset of $\mathbf{P}/\operatorname{poly}$ and commonly believed that $\mathbf{TC}^0$ is a strict subset of $\mathbf{P}/\operatorname{poly}$, though the strictness is still an open problem to be proved. As a derivation, T-LLMs, which are $\mathbf{TC}^0$ circuits, cannot be general learners, which are at least $\mathbf{P}/\operatorname{poly}$-complete.

**Theorem 3.** *Assume $\mathbf{TC}^0 \neq \mathbf{P}/\operatorname{poly}$. Suppose that $\mathcal{F}_0 = B_2$ and $\mathcal{T}$ denotes the set of threshold gates. There exists a polynomial $s$ such that for any integer $d_0$ and polynomials $l(\cdot), s_0(\cdot)$, there exists an integer $n$ and such that no circuit in $\Gamma(n + l(n), 1, s_0(n), d_0, \mathcal{F}_0 \cup \mathcal{T})$ is a universal circuit for $\Gamma(n, 1, s(n), \infty, \mathcal{F}_0)$.*

**Corollary 2.** *Assume $\mathbf{TC}^0 \neq \mathbf{P}/\operatorname{poly}$. For any integer $d_0$ and polynomial $s_0(\cdot)$, there exists an integer $n$ such that no log-precision Transformer-based LLM of depth $d_0$ and size $s_0(n)$ can faithfully follow all the prompts of complexity $n$.*

In the above analysis, we adopt an intuition that prompts describe instructions to guide the inference of LLMs. It seems that our analysis only considers the prompts that provide direct instructions and neglects the important case of in-context learning with few-shot demonstrations. This is not true. Under the universal circuit perspective, the few-shot demonstrations can also be seen as circuit encodings and our analysis applies agnostic to the types of the encodings. In fact, we can further more show that as circuit encodings, the few-shot demonstrations cannot be more efficient than

standard circuit encodings. While the standard circuit encodings for circuits of size $s$ take $O(s \log s)$ bits (Cook & Hoover, 1985), a lower bound of encoding by demonstration is $\Omega(s \log s)$ bits. We state this lower bound in Theorem 4.

**Theorem 4.** *Suppose that $\mathcal{F}_0 = B_2$. It takes at least $\Omega(s \log s)$ bits to encode all the circuits in $\Gamma(n, 1, s, \infty, \mathcal{F}_0)$ by demonstration.*

## 5 INCONSISTENCY: THEORETICAL FAILURE AND EMPIRICAL SUCCESS

Our theoretical results indicate that T-LLMs fail to be general learners. However, the T-LLMs achieve great empirical success in various tasks. We provide a possible explanation for this inconsistency: while T-LLMs are not general learners, they can partially solve complex tasks by memorizing a number of instances, leading to an illusion that the T-LLMs have genuine problem-solving ability for these tasks. Theorem 5 shows that a $\mathbf{TC}^0$ circuit is able to memorize a polynomial number of instances for any problem.

Another evidence for this explanation is that T-LLMs have inconsistent performance on different tasks. One confusing phenomenon is: on the one hand, they may solve complex tasks such as coding and writing, even outperforming human experts in some cases; on the other hand, they can still make mistakes on simple or even seemingly trivial problems such as simple arithmetic and simple symbolic manipulation. The negative aspect can be partially explained by our result that T-LLMs are not general learners: circuits that generally solve these simple tasks are beyond the circuits that can be simulated by the universal circuits of the T-LLMs. For the positive aspect, one possibility of T-LLMs' success in solving certain complex tasks is that they memorize some instances of problems and answers.

**Theorem 5.** *Suppose that $\mathcal{F}_0 \in B_2$ and $\mathcal{T}$ denotes the set of threshold gates. For any integer $n$, polynomial $q(\cdot)$ and $\mathcal{I}_n : \mathcal{F} \mapsto 2^{\Sigma^n}$ such that $|\mathcal{I}_n(f)| \leq q(n)$ for all $f \in B_n$ and each $f \in B_n$, there exists a circuit $\gamma_f \in \Gamma(n, 1, s_0(n), d_0, \mathcal{F}_0 \cup \mathcal{T})$ such that $\gamma_f(x) = f(x)$ for all $x \in \mathcal{I}_n(f)$.*

## 6 RELATED WORK

**Capabilities of LLMs.** The emergent abilities of LLMs in solving a wide variety of tasks are one of their most impressive features. Extensive works have made attempts to evaluate different aspects of LLMs' capabilities (Wei et al., 2022; Bubeck et al., 2023; Saparov et al., 2023; Chen et al., 2021; Zhang et al., 2022; Frieder et al., 2023; Webb et al., 2023; Sun et al., 2023; Wang et al., 2023b; Jin et al., 2023) and probe the mechanism behind (Saparov & He, 2022; Wang et al., 2023a; Hao et al., 2023; Tang et al., 2023; Prystawski & Goodman, 2023; Radhakrishnan et al., 2023; Pezeshkpour & Hruschka, 2023), outlining the boundary of the LLMs' capabilities. These studies provide important insights into what LLMs can or cannot do and why. Motivated by these results, we propose a paradigm to characterize the capabilities in terms of universal circuits. Following the paradigm, we show that the capabilities of realistic T-LLMs are not as strong as general learners.

**Computational Models for LLMs.** To investigate and analyze LLMs, there have been attempts to propose computation models for LLMs, as proper alternatives that capture some significant aspects of the extremely complex systems. A line of works explains in-context learning by viewing LLMs as computational models such as implicit Bayesian inference (Xie et al., 2021), internal meta-optimization Von Oswald et al. (2023); Dai et al. (2023), kernel regression Han et al. (2023), etc. Delétang et al. (2023) take a compression perspective of language models to demonstrate its predictive capabilities. Dziri et al. (2023) considers compositional tasks as computation graphs and multi-step reasoning as linearized subgraph matching, depicting LLMs' limits on compositionality. Our work abstracts LLMs as universal circuits. Following this abstraction, we reduce the problem of whether LLMs are general reasoners to the problem of whether certain universal circuits can be implemented by the circuit family of the realistic models.

**Computational Power of Transformers.** Transformers (Vaswani et al., 2017) are one of the prevailing architectures in machine learning and also the current paradigm of LLMs. Some researchers have adopted perspectives from theoretical computer science to reveal the computational power of Transformers. For ideal Transformers without limitations on precision, depth, and size, they are proved to be Turing complete (Pérez et al., 2019; 2021), i.e., they can solve all computable prob-

lems in theory. However, when more restrictions under realistic settings are considered, there are striking gaps between the ideal Transformers and Transformers in practice. Hahn (2020); Hao et al. (2022) show that hard attention Transformers are within the complexity class $\mathbf{AC}^0$ and cannot recognize non-$\mathbf{AC}^0$ formal language. Merrill et al. (2022); Merrill & Sabharwal (2023) prove that log-precision Transformers with constant depths and polynomial sizes belong to the complexity class $\mathbf{TC}^0$. Our work builds a framework that describes a relation between the computational power of Transformers and the capabilities of T-LLMs in the language of universal circuits. Under the framework, we show that T-LLMs are not general learners, as a result of realistic Transformers being within $\mathbf{TC}^0$ (Merrill et al., 2022; Merrill & Sabharwal, 2023).

Feng et al. (2023) shows that CoT can help T-LLMs to $\mathbf{NC}^1$-complete problems, which is beyond the model capacity under the assumption that $\mathbf{TC}^0 \subsetneq \mathbf{NC}^1$. Although their conclusion appears to be contradictory to our analysis, the two results are actually not in conflict because the settings are different. We first need to distinguish two types of prompt engineering techniques that are both discussed in the literature with the name CoT but of fundamental differences. One type is to describe a step-by-step problem-solving method within one prompt, either directly or via demonstration, to instruct the LLMs to perform step-by-step inference. The other type is to break the solving process of the original problem into subproblems and ask the LLMs to solve one subproblem in each query; the whole solving process of this type involves repeated and recursive queries to the LLMs. We refer to the first type as Type I and the second type as Type II[1]. Our analysis of T-LLMs applies to Type I CoT. The general problem-solving capabilities of T-LLMs are still restricted by the expressive capacity of the realistic Transformers and thus no Type I CoT can generally solve problems beyond $\mathbf{TC}^0$. Other works such as (Feng et al., 2023) show that T-LLMs with Type II CoT can break the model capacity limitations and solve some problems beyond $\mathbf{TC}^0$ by repeatedly utilizing T-LLMs for each step. The combination of results from both aspects suggests a trade-off to be considered in the prompt design: the complexity of subproblem decomposition (by human intelligence) and the complexity of subproblems (by LLMs).

## 7 CONCLUSION AND DISCUSSION

### 7.1 CONCLUSION

In this work, we analyze the capabilities of T-LLMs on general problem solving from the perspective of universal circuits. Under the framework, we derive a necessary condition for LLMs being general learners is that the expressive power of realistic T-LLM model class can cover universal circuits for all circuits of polynomial size, which is at least as hard as $\mathbf{P}/\operatorname{poly}$-complete. With this reduction, the result that realistic T-LLMs are within $\mathbf{TC}^0$, and the assumption that $\mathbf{TC}^0 \neq \mathbf{P}/\operatorname{poly}$, we conclude that T-LLMs are not general learners.

### 7.2 LIMITATIONS

Our universal circuit framework is based on the ideal setting that prompts provide "programs" and LLMs follow the prompts faithful to solve tasks. There are deviations that LLMs may solve problems with inaccurate prompts. Our framework does not cover the explanations for these cases.

Our work only considers T-LLMs from the perspective of their representation power, which is insufficient to capture everything of T-LLMs' limitations. Besides the representation power, there are many other important aspects to be considered to have a more precise characterization of T-LLMs' limitations, e.g., data, learning, optimization, randomness, etc. These aspects are not included in the framework of this work.

From the viewpoint of circuit complexity theory, it is a relatively "loose" upper bound that T-LLMs are not general learners. This upper bound roots from the assumption that $\mathbf{TC}^0 \subsetneq \mathbf{P}/\operatorname{poly}$. A common belief in circuit complexity theory, which has not been strictly proved though, is that there are still many complexity classes between $\mathbf{TC}^0$ and $\mathbf{P}/\operatorname{poly}$. Our result does not exploit this hierarchy and thus only serves as a rather "loose" upper bound.

---

[1]Some researchers (Radhakrishnan et al., 2023) also refer Type II CoT as Question Decomposition (QS) to distinguish and emphasis the difference.

### 7.3 FUTURE DIRECTIONS

**Innovative architecture designs beyond Transformers.** Our work shows that T-LLMs cannot be general learners because of the inherent limitations on the power of realistic Transformers. To develop general learners with foundation models, we require innovative architectures that have more representation capacities. Designing such architectures beyond the current paradigms of Transformers is an important direction for future research.

**Finer-grained analysis on LLM capabilities from the universal circuit perspective.** The proposed universal circuit framework enables a systematic analysis of various LLMs. It captures the core elements of LLMs' inference processes through the lens of universal computation. To assess whether T-LLMs are general learners, we employ a three-step paradigm: 1) Identify the circuit sets to be simulated by universal circuits for targeted LLM capabilities; 2) Determine the models' complexity class; 3) Check whether the universal circuits are implementable by the models' complexity class. The paradigm reduces the analysis on LLM capabilities to the analysis on the existence of universal circuits, allowing us to borrow the tools from circuit complexity theory. This versatile framework lays the groundwork for future studies. We plan to extend our analysis to further explore T-LLMs' capabilities and to investigate LLMs with different architectures, thereby refining the relationship between model capabilities and architectures.

**Principles for prompt engineering.** Our analysis shows that for a single query to an LLM, the LLM capabilities are always limited by the underlying model regardless of prompts. A promising future direction is to develop principles for prompt engineering to compose multi-queries such as Type II CoT (Radhakrishnan et al., 2023; Feng et al., 2023). By properly composing multi-queries, we may expect the system to break the limitations due to the model capacity, striking a balance between the complexity of multi-query composition and the complexity of a single query. Developing theories and algorithms for such prompt engineering techniques can help people to better utilize the LLMs.

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

# A    PROOFS

## A.1    PROOF OF THEOREM 1

*Proof.* Theorem 1 is a variant of the classic Shannon's lower bound (Shannon, 1949) on the size of "most" circuits to compute Boolean functions. We adopt a similar counting argument to show no circuit in $\Gamma(p(\log n) + l(n), 1, s_0(n), \infty, \mathcal{F}_0)$ is a universal circuit for $B_{p(\log n)}$.

Assume that for any polynomial $p(\cdot)$ and integer $n$, there exists a circuit $\alpha \in \Gamma(p(\log n) + l(n), 1, s_0(n), \infty, \mathcal{F}_0)$ that is a universal circuit for $B_{p(\log n)}$, where $l(\cdot)$ and $s_0(\cdot)$ are some polynomials. For each circuit encoding $e_f \in \Sigma^{l(n)}$, $\alpha(\cdot, e_f)$ corresponds to a circuit in $\Gamma(p(\log n), 1, s_0(n), \infty, \mathcal{F}_0)$. Under the assumption that $\alpha$ is a universal circuit for $B_{p(\log n)}$, there are at least $\left|B_{p(\log n)}\right|$ different circuits in $\Gamma(p(\log n), 1, s_0(n), \infty, \mathcal{F}_0)$. Denote $N(m, s)$ be the number of different circuits in $\Gamma(m, 1, s, \infty, \mathcal{F}_0)$. We have

$$N(m,s) \overset{(a)}{\leq} \frac{s\left[|B_2|\,(m+s)^2\right]^s}{s!} = \frac{s\left[16(m+s)^2\right]^s}{s!} \overset{(b)}{\leq} \frac{s\left[16e(m+s)^2\right]^s}{es^s},$$

where the inequality $(a)$ is obtained by counting different circuits in $\Gamma(m, 1, s, \infty, \mathcal{F}_0)$ and the inequality $(b)$ comes from the inequality $s! \geq e\left(\frac{e}{s}\right)^s$ derived from Stirling's formula.

Taking the logarithm of both sides, we have

$$\log N(m,s) \leq \log s + s\log\left[16(m+s)^2\right] - \log e - s\log s$$
$$\leq \log s + s\log\left[16\left(1 + \frac{m}{s}\right)(m+s)\right].$$

Noting that $|B_m| = 2^{2^m}$ and plugging $m = p(\log n)$ and $s = s_0(n)$, we have

$$\frac{\log N(p(\log n), s_0(n))}{\log |B_{p(\log n)}|} \leq \frac{\log s_0(n) + s_0(n)\left[16\left(1 + \frac{p(\log n)}{s_0(n)}\right)(p(\log n) + s_0(n))\right]}{2^{p(\log n)}}.$$

For $p(x) = x^2$ and $n \to \infty$, we have $\frac{\log N(p(\log n), s_0(n))}{\log |B_{p(\log n)}|} \to 0$, This result implies that $\alpha \in \Gamma(p(\log n) + l(n), 1, s_0(n), \infty, \mathcal{F}_0)$ cannot be a universal circuit for $B_{p(\log n)}$ for sufficiently large $n$, contradicting the assumption. Therefore, no circuit in $\Gamma(p(\log n) + l(n), 1, s_0(n), \infty, \mathcal{F}_0)$ is a universal circuit for $B_{p(\log n)}$. $\square$

## A.2    PROOF OF THEOREM 3

*Proof.* We prove Theorem 3 by contradiction. Assume that for any polynomial $s$, there exist an integer $d_0$ and polynomials $l(\cdot), s_0(\cdot)$ such that a circuit $\alpha \in \Gamma(n + l(n), 1, s_0(n), d_0, \mathcal{F}_0 \cup \mathcal{T})$ is a universal circuit for $\Gamma(n, 1, s(n), \infty, \mathcal{F}_0)$.

According to the assumption, for each $\gamma \in \Gamma(n, 1, s(n), \infty, \mathcal{F}_0)$, there exists a circuit encoding $e_f \in \Sigma^{l(n)}$ such that $\alpha(x, e_f) = \gamma(x)$ for all $x \in \Sigma^n$. For each circuit encoding $e_f \in \Sigma^{l(n)}$, $\alpha(x, \cdot)$ implements a circuit in $\Gamma(n, 1, s_0(n), d_0, \mathcal{F}_0 \cup \mathcal{T})$. Therefore, for any polynomial $s$, there exist an integer $d_0$ and a polynomial $s_0(\cdot)$ such that any circuit in $\Gamma(n, 1, s(n), \infty, \mathcal{F}_0)$ can be simulated by a circuit in $\Gamma(n, 1, s_0(n), d_0, \mathcal{F}_0 \cup \mathcal{T})$.

Under the assumption that $\mathbf{TC}^0 \neq \mathbf{P}/\text{poly}$, there exist circuits $\{\gamma_n\}$, $\gamma_n \in \Gamma(n, 1, s(n), \infty, \mathcal{F}_0)$ for some polynomial $s(\cdot)$ such that for any polynomial $s_0(\cdot)$, no circuits $\{\gamma'_n\}$, $\gamma'_n \in \Gamma(n, 1, s_0(n), d_0, \mathcal{F}_0 \cup \mathcal{T})$ can simulate $\{\gamma_n\}$, i.e., $\gamma'_n(x) = \gamma_n(x)$ for all $x \in \Sigma^n$, $n = 1, 2, \ldots$. This contradicts the previous inference drawn from the existence of the universal circuit. $\square$

### A.3 PROOF OF THEOREM 4

*Proof.* We begin the proof by a lemma on the number of different functions that the set of all the $n$-input-$s$-size circuits can represent.

**Lemma 1.** $\Gamma(n, 1, s, \infty, B_2)$ *can be divided into at least* $\Omega(s^s)$ *equivalent classes.*

*Proof of Lemma 1.* We derive the lemma from Lupanov's upper bound on the circuit size to compute all Boolean functions of $n$ input. We follow the description of the theorem in Sarma (2012).

**Theorem 6** (Lupanov's Upper Bound (Lupanov, 1958))**.** *A function on input defined on the complete basis* $\Omega = \{\neg, \vee, \wedge\}$ *of two input gates can be computed by a circuit of size* $(1 + o(1))\frac{2^n}{n}$.

According to Theorem 6, there exists a constant $c$ such that for all integer $n$, any Boolean functions of $n$ input can be computed by a circuit in $\Gamma\left(n, 1, c\frac{2^n}{n}, \infty, B_2\right)$. This implies that for $n_0 \leq n$, any Boolean functions of $n_0$ input can be computed by a circuit in $\Gamma(n, 1, s, \infty, B_2)$ (for $n > n_0$, consider the first $n_0$ input bits). Since different Boolean functions correspond to different equivalent classes, there are at least $|B_{n_0}| = 2^{2^{n_0}}$ equivalent classes in $\Gamma(n, 1, s, \infty, B_2)$.

Let $n_0^*$ be the maximal $n_0$ such that all the functions in $B_{n_0}$ can be computed by $\Gamma(n, 1, s, \infty, B_2)$. Then we have

$$2c\frac{2^{n_0^*}}{n_0^*} \geq s. \tag{1}$$

Otherwise, we have

$$c\frac{2^{n_0^*+1}}{n_0^*+1} \leq 2c\frac{2^{n_0^*}}{n_0^*} < s,$$

which implies all the functions in $B_{n_0^*+1}$ can be computed by $\Gamma(n, 1, s, \infty, B_2)$, contradicting the maximality of $n_0^*$. By equation 1, we have $2^{n_0^*} \geq \frac{n_0^* s}{2c}$ and $n_0^* \geq \log\frac{s}{2c}$. Hence, we can further have

$$\left|B_{n_0^*}\right| = 2^{2^{n_0^*}} \geq 2^{\frac{n_0^* s}{2c}} \geq 2^{\frac{s\log\frac{s}{2c}}{2c}} = \Omega(s^s).$$

$\square$

We are now ready to prove Theorem 4. Let $M(n, k)$ be the number of equivalent classes that can be distinguished by $k$ $n$-input instances. We have

$$M(n, k) \leq \binom{2^n}{k} 2^k.$$

By Lemma 1, there are at least $O(2^s)$ equivalent classes. Hence, to encode all the circuit in $\Gamma(n, 1, s, \infty, \mathcal{F}_0)$ with demonstration of $k$ $n$-input instances, we need at least distinguish $\Omega(s^s)$, i.e.,

$$M(n, k) \geq cs^s, \tag{2}$$

for some constant $c$. As a result, we require

$$\binom{2^n}{k} 2^k \geq cs^s.$$

Plugging $\binom{a}{b} \leq \left(\frac{ea}{b}\right)^b$ with $a = 2^n$ and $b = k$ and taking the logarithm of both sides, we have

$$k \log \frac{e \cdot 2^n}{k} + k \log 2 \geq s \log s + \log c.$$

With $\frac{e}{k} \leq 4$, we can further simplify the inequality and obtain

$$k \geq \frac{s \log s + \log c}{n + 2 + \log 2}.$$

Since each instance takes $n + 1$ bits, it requires at least

$$(n + 1)k \geq (n + 1)\frac{s \log s + \log c}{n + 2 + \log 2} = \Omega(s \log s)$$

$\square$

### A.4 PROOF OF THEOREM 5

*Proof.* For each $n$ and $f \in B_n$ we construct a circuit $\gamma_f$ of depth 3 and size $(n + 1)q(n) + 1$ such that $\gamma_f(x) = f(x)$ for all $x \in \mathcal{I}_n(f)$.

We first construct a function $m_{x_0}(\cdot)$ by a circuit of depth 2 and size $n + 1$ such that $m_{x_0}(x) = 1$ if and only if $x = x_0$.

**Lemma 2.** *For each $x_0 \in \Sigma^n$, there exists a circuit of depth 2 and size $n + 1$ such that simulates the function $m_{x_0}(\cdot)$.*

*Proof of Lemma 2.* Suppose that $x_0 = x_0^{(1)}x_0^{(2)} \cdots x_0^{(n)}$ where $x_0^{(i)} \in \Sigma$ for $i = 1, 2, \ldots, n$. We construct a desired circuit as follows (see Figure 3a):

- There are $n$ gates $g_1, g_2, \ldots, g_n$ such that $g_i$ is IDENTITY if $x_0^{(i)} = 1$ and is NOT otherwise.

- There are links from $x_0^{(i)}$ to $g_i$ for all $i = 1, 2, \ldots, n$.

- There is a threshold gate $T_n$ taking $n$ inputs such that $T_n$ outputs 1 if the sum of the inputs is greater or equal to $n$ and outputs 0 otherwise.

- There are links from $g_i$ to $T_n$ for all $i = 1, 2, \ldots, n$.

- $T_n$ is the output gate.

By the construction, $T_n(x)$ outputs 1 if and only if all the bits of $x$ are the same as $x_0$, i.e., $x = x_0$. $\square$

We then construct a circuit $\gamma_f$ such that $\gamma_f(x) = f(x)$ for all $x \in \mathcal{I}_n(f)$. Without loss of generality, suppose that $\mathcal{I}_n(f, 1) = \{x \in \mathcal{I}_n(f) \mid f(x) = 1\} = \{x_1, x_2, \ldots, x_k\}$ and $k \geq 1$ (if $m = 0$, we can trivially construct $\gamma_f$ that always output 0). The circuit $\gamma$ is constructed as follows (see Figure 3b):

- There are $k$ blocks $m_{x_1}, m_{x_2}, \ldots, m_{x_k}$.

- There are links from the input $x$ to $m_{x_i}$ for all $i = 1, 2, \ldots, k$.

- There is a threshold gate $T_1$ taking $n$ inputs such that $T_1$ outputs 1 if the sum of the inputs is greater or equal to 1 and outputs 0 otherwise.

- There are links from $m_{x_i}$ to $T_n$ for all $i = 1, 2, \ldots, n$.

- $T_1$ is the output gate.

By the construction, $\gamma_f$ outputs 1 if and only if the input equals some $x_i \in \mathcal{I}_n(f, 1)$, which implies that $\gamma_f(x) = f(x)$ for all $x \in \mathcal{I}_n(f)$. Since each block $m_{x_i}$ contains $n + 1$ gates and is of depth 2 and there are at most $q(n)$ such blocks, the circuit $\gamma_f$ is of depth 3 and size $(n + 1)q(n) + 1$.

$\square$

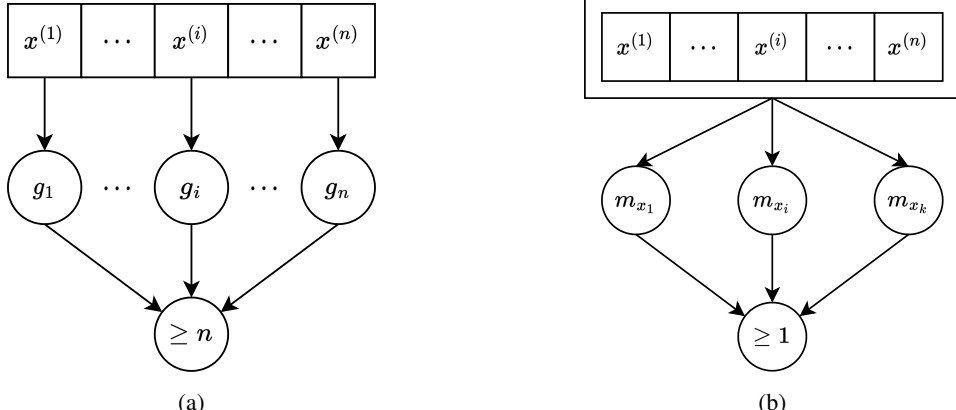

Figure 3: (a) Illustration of the circuit that simulates $m_{x_0}$. (b) Illustration of the circuit that outputs 1 on $\{x_1, x_2 \ldots, x_k\}$ and 0 otherwise.

