# OpenReview forum: "Transformer-Based Large Language Models Are Not General Learners: A Universal Circuit Perspective"
_ICLR.cc/2024/Conference — Submitted to ICLR 2024_

### Official Review · Reviewer_PAUt · 2023-10-25

**Soundness:** 3 good
**Presentation:** 3 good
**Contribution:** 2 fair
**Rating:** 5
**Confidence:** 4

**Summary:**

The paper studies the limitation of Transformers in expressing complex function, arguing that Transformer-based LLMs are limited in their capacity to represent universal circuits. This result leverages previous results showing that Transformers of constant depth can only represent functions in $TC^0$. To explain the gap between the practical success of Transformers and their theoretical limitations, the authors prove that Transformers are able to memorize O(poly(n)) instances.

**Strengths:**

The discussion on the implication of the limitations of Transformers in only expressing functions in $TC^0$ is interesting. Studying the actual limitations of Transformers in terms of their capabilities, and not just in terms of function class approximation, is important.

**Weaknesses:**

While the premise of the paper seemed interesting, I find that the paper does not give a proper discussion of some relevant aspects of Transformer-based LLMs.

First, the discussion on the effect of using Chain-of-Thought (CoT) and the relation to the negative result is not clear. The authors do acknowledge that CoT with auto-regressive inference allows Transformers to compute functions beyond $TC^0$, but claim that this somehow does not apply to their setting. The authors need to clarify whether their results cover Transformer used for decoder-like auto-regressive inference, where in each iteration the Transformer is used to generate the next token. If so, why does CoT not solve the limitation of computing functions in $TC^0$? While every iteration is "in $TC^0$", the overall function computed by the Transformer can go beyond this limitation. If this is because the Transformer cannot output tokens that are "ignored" during the computation, I believe this does not capture properly the behavior of Transformer-based LLMs, which clearly can utilize CoT in a way that overcomes this limitation.

Second, it is not clear which results are novel contributions (on the technical level), and which results are just "rehashing" of known results. As the limitation of Transformers in only expressing functions in $TC^0$ has already been established, what results are novel to the paper? Is the analysis of limitations of $TC^0$ functions in expressing universal circuits new? I think that the authors should clarify whether they are just stating known facts about the limitation of $TC^0$, whether they are proving new results on the capabilities of $TC^0$ functions, or whether the setting of language-modeling with $TC^0$ generates new theoretical results that are not already studied in the classical circuit complexity literature.

Some minor comments:
- I believe that the term "general learner" is confusing, as the discussion in not about what functions can be learned by Transformers (i.e., given training samples), but rather what functions they can approximate.
- The claim that "LLMs can solve complex tasks by memorizing some instances" is somewhat problematic. The fact that Transformers are able to memorize inputs does imply that this is how complex tasks are solved.
- The beginning of Remark 1 is confusing, and should be rewritten.

**Questions:**

See above.

---

### Official Review · Reviewer_PfJ8 · 2023-10-31

**Soundness:** 3 good
**Presentation:** 2 fair
**Contribution:** 3 good
**Rating:** 5
**Confidence:** 4

**Summary:**

This paper considers transformers' ability to execute instructions (follow prompts) written as circuits. The main result is that

**Strengths:**

1. I really like the research question and goals of this work, as it draws an interesting connection between transformers' ability to execute instructions (posed as circuits) and existing results analyzing transformers via circuits and classical work on universal circuits.
2. The work successfully leverages the framing of general learners in terms of circuits to conclude that transformer LMs are not universal learners.

**Weaknesses:**

If the authors can adequately address the following issues around presentation and the interpretation of Theorem 5, I would be satisfied raising my score, as I value the research question raised by the paper and believe the conclusions are valid.

## "General Learner": Missing Formal Definition and Misleading Name

The "general learner" concept used in the title and throughout is named in a somewhat misleading way, as the results here have to do more with *expressive power* than learning. That is, by "general learner" the authors mean a model that can *express* a universal circuit. This is a bit confusing and I would suggest either changing the terminology or explicitly clarifying that you are talking about expressive power and not learning (which is fine, limitations on expressive power translate to limitations on what can be learned).

More of an issue is that the fact that the definition of general learner in Section 2.2 is not clearly structured. After reading it several times, I get the idea that the authors view a model as a general learner if it can simulate a universal circuit for all poly-size circuits. Yet it seems inconsistent at times whether this is a formal definition of the concept or a necessary condition obtained from some other (unprovided) definition, as Proposition 1 suggests.
I would suggest rewriting Section 2.2 in one of two ways:
1. Provide a formal definition for "general learner" before stating the proposition.
2. Leave "general learner" imprecise but reframe proposition 1 as your formal definition attempting to capture it: you view an LM hypothesis class is a general learner if it can express a universal circuit family for all poly-size circuits. You can then argue informally for why you think this definition makes sense, akin to, e.g., the Church-Turing thesis.

## (Minor) Imprecise Claim about Poly(n) Size

In Theorem 1, the authors claim:
> We consider log-precision, constant-depth, and polynomial-size Transformers: for Transformers whose input is of length n, the values at all neurons are represented with O(log n) bits, the depth is constant, and the number of neurons is O(poly (n)). This setting captures the design and the hardware restriction of realistic T-LLMs and is common in the literature on the theoretical power of T-LLMs (Hahn, 2020; Hao et al., 2022; Merrill et al., 2022; Merrill & Sabharwal, 2023).

This is almost correct, but largely these papers consider a slightly different setting where the number of neurons is O(1), and all other complexity measures are as described. However, it should be straightforward to extend the results from these papers to the poly(n) case, at least in the nonuniform setting considered here. It would be good to make this more explicit here though.

## Logical Flow of Main Results and Relation to Prior Work Should be Clarified

The paper defines T-LLMs as general learners if "the expressive power of realistic T-LLM model class can cover universal circuits for
all circuits of polynomial size" and says that:

> It is unknown whether the realistic Transformers are expressive enough to be general learners

In fact, it seems that Theorem 3 essentially follows from the TC0 upper bound in previous work. Here's how I would reconstruct the proof of Theorem 3:

*Proof.* Assume T-LLMs are general learners by contradiction. Take any P-complete problem p. p has a poly-size circuit by membership in P, and thus T-LLMs must be able to execute some circuit to solve p. But then there is a threshold circuit with input size n + poly(n), constant depth, and size poly(n + poly(n)) that solves p. This means that p is in TC0, which results in contradiction unless TC0 = P (conjectured false).

Thus, it seems to me that you are essentially applying past results to answer a specific question you have (which is still a valuable contribution). However, it is confusing to me how this interacts with the prior results in the paper. How does Theorem 2 come into play when proving Theorem 3, if Theorem 3 can be proved independently? Is Theorem 2 a stronger version in some sense because it applies for polylog input size?

Another question: does "prompts of complexity n" mean prompts of length n? If so, does this follow from Theorem 2? This isn't clear from the current structure if so, since it's stated as a corollary of Theorem 3.
In general, it may help to reorder some of these results, add forward references to the proofs, and indicate how different results depend on one another.

## Claims about Instance Memorization are Very Speculative

The authors show that a poly(n)-size transformer can in principle memorize poly(n) instances, but it is quite a jump to the idea that this instance-based memorization is what explains the success of T-LLMs in practice.
There is a lot missing to actually justify this claim:
1. For one, it would require doing some experiments with trained LMs and finding evidence of memorization. In fact, empirical evidence suggest that LMs do memorize n-grams from their training data somewhat, but not full examples (see [McCoy et al.](https://arxiv.org/abs/2111.09509))
2. There are theoretical reasons why it would be difficult for a T-LLM to learn to memorize instances from single examples. T-LLMs are trained with huge batches, and it can be hard to pick out all the information about one example from a batch. This idea has been used to show lower bounds on the learnability of functions like sparse parities for feedforward neural networks (see [Barak et al.](https://arxiv.org/abs/2207.08799))

The authors have already hedged the claim that memorization is responsible for the success of T-LLMs, but I think it could be toned down further and, if the authors would like to engage in speculation, it would be good to integrate some of this related work and its implications into the discussion (e.g., memorizing full instances from a single occurrence doesn't seem to happen in practice and there are theoretical arguments against it).

**Questions:**

Do we really want models that can express universal circuits? Perhaps constraints on the expressive power of the hypothesis class could be useful for the inductive bias of models. It could be worth discussing the potential benefits of having a constrained hypothesis class in the limitations section.

## Related Work

* Finlayson et al. benchmark transformer's ability to execute regular expressions (rather than circuits): https://www.semanticscholar.org/paper/What-Makes-Instruction-Learning-Hard-An-and-a-New-a-Finlayson-Richardson/cb16b85891172572cd856142880b503db0c2bc61

---

> ### Author Response · Authors · 2023-11-18
>
> Thank you for your insight comments.
>
> - *Claims about Instance Memorization are Very Speculative*
>
> Thank you for suggestion. Our current claim is that from the perspective of expressive, it is possible that the model could achieve something by memorization. We agree that it requires more discussion on the learnability and the training dynamics to further investigate whether learning to memorize is feasible and accounts for the success of LLMs (at least for some problems).
>
> - *Do we really want models that can express universal circuits? Perhaps constraints on the expressive power of the hypothesis class could be useful for the inductive bias of models. It could be worth discussing the potential benefits of having a constrained hypothesis class in the limitations section.*
>
> We think that constraints on the expressive power of the hypothesis class may benefit the learning of the model. Some classic results in learning theory  (e.g., VC dimension) have shown that a limited hypothesis class is important for learnability. Despite that these results may not directly apply to the LLMs, we think the idea that a limited hypothesis class  benefits learnability could be insightful. We believe it is an interseting direction for future work to reveal the underlying mechanism (if exists) how the idea works for LLMs.

---

> > ### Comment · Reviewer_PfJ8 · 2023-11-22
> >
> > Thanks for your response. I maintain my current score.
> >
> > I like the idea of this paper and encourage the authors to further address the other points above ("Logical Flow of Main Results" and "General Learner"), as well as the other reviewers' suggestions, which could make the paper's contributions clearer and stronger.

---

### Official Review · Reviewer_joMD · 2023-11-03

**Soundness:** 3 good
**Presentation:** 2 fair
**Contribution:** 2 fair
**Rating:** 3
**Confidence:** 2

**Summary:**

This paper considers the question of what problems LLMs can solve, by studying their expressive power. In particular the paper shows that  transformer based LLMs (TLLMs) cannot be general problem solvers, as they fall in a class of circuits TC0, which is known to only solve problems from a limited class. The key contribution of the paper is a characterisation of a TLLM as a universal circuit, which is a circuit that has the ability to take a function description (f) and a function input (x), and return f(x) for any function and any (x). The paper then shows that the current design of transformers cannot possibly lead to a universal circuit due to the structural limitations of transformers.

**Strengths:**

--

**Weaknesses:**

I had trouble following the paper as the definitions and claims have not been presented well. It is really hard to find the main technical claim of the paper, and the introduction was of little help. The paper essentially argues that not having the ability to solve \textit{all} problems, raises doubts on LLM capabilities. It is not clear why it can't just be \textit{most} problems, or most instances of most problems.


Although the paper reduces the problem to a well known results, it can still be worthwhile to provide a single example of a problem that cannot be solved to get an idea of what falls outside the purview of current models.

**Questions:**

Randomness seems to be a key part of the success of modern LLMs, and it is not considered at all in the paper. Is the expressive power of TLLM circuit more if it is provided with additional random input. Moreover, the TLLMs can be thought of as providing a list of answers (as the output is a real vector). Does this ability add any power to the circuit class?

---

### Official Review · Reviewer_T96k · 2023-11-05

**Soundness:** 2 fair
**Presentation:** 1 poor
**Contribution:** 2 fair
**Rating:** 6
**Confidence:** 3

**Summary:**

This paper presents an intriguing approach to enhance graph transformers by incorporating positional encodings derived from quantum correlations, motivated by the capabilities of quantum processing units. While the concept is innovative, there are several areas where the paper could benefit from additional clarity and justification.

**Strengths:**

This work has the potential to set a precedent in the fusion of quantum computing with graph transformers. Enhancing the paper with the aforementioned suggestions could substantially improve its impact and reception by the research community.

**Weaknesses:**

Definition of Variables and Positional Encodings:

The paper would greatly benefit from a more detailed explanation of the positional encodings used. Terms such as θ, t, δ, and especially the adjacency matrix A, which are crucial for understanding the method, require clear definitions and contextual usage within the proposed quantum framework.
Furthermore, an explicit definition of the feature matrix X of the nodes would help in understanding how these features interact with the quantum-inspired positional encodings.
Clarity on Quantum Enhancements:

Section 3.2.2 seems to lack depth in the explanation of how the quantum correlations are calculated and utilized within the graph transformers. Providing a more comprehensive elucidation of these processes would aid in bridging the gap between classical and quantum approaches.
A layman's explanation or intuitive reasoning behind the adoption of quantum features and their computational advantages in graph analysis would be invaluable for the reader's understanding.
Theoretical Advantages and Theorem Justification:

While you mention that quantum features are theoretically more expressive, the paper falls short of explaining the underlying intuition and proof for this assertion. Elaboration on Theorem 1, with an intuitive breakdown of its implications, would significantly enhance the readability and credibility of the results.
Comparative Analysis and Benchmarking:

The comparisons presented in Tables 1 and 2 focus solely on the improvements over one reference work (Ma et al., 2023). For a more comprehensive analysis, it would be instructive to see how the approach compares with a broader spectrum of state-of-the-art methods.
Additionally, the rationale for choosing (Ma et al., 2023) as a benchmark, along with the significance of the improvements observed, even if minute, should be clearly articulated. It's important to discuss why these improvements are non-trivial and how they advance the field, considering the rapidly evolving landscape of both quantum computing and graph analysis.

**Questions:**

In Section 3.2.2,  could you expand on the methodology for calculating and using these correlations? Any intuitive insights into this process would also be greatly appreciated.

---

### Official Review · Reviewer_8H3T · 2023-11-08

**Soundness:** 4 excellent
**Presentation:** 2 fair
**Contribution:** 1 poor
**Rating:** 1
**Confidence:** 5

**Summary:**

This paper shows that realistic transformers (1) cannot represent any function on polylog(n) bits, and (2) cannot represent any polynomial-time computable circuit.

**Strengths:**

The question about whether realistic transformers can represent any polynomial-time-computable function is interesting.

**Weaknesses:**

All of the results in this work seem to be previously known:
* Theorem 1 is general to any polynomial-size circuit -- there is nothing special about Transformers. It proves that not all functions are polynomial-time computable via a counting argument, which is previously known.
* Theorem 2 is a restatement of past work, showing that transformers lie in logspace-uniform TC^0
* Theorem 3 assumes TC^0 \neq P / poly, and then derives that transformers cannot simulate any poly-time circuit. This is immediate from Theorem 2, and this kind of separation appears to have been the implicit the point of Merrill and Sabharwal 2023

* This statement seems overly strong and minimizes prior work: "This work takes the first step towards rigorously answering the fundamental question by considering a theoretical boundary of model capacity to be general learner"
    * e.g., "Saturated Transformers are Constant-Depth Threshold Circuits" shows that transformers lie in TC^0 under a saturation condition
    * e.g., "The Parallelism Tradeoff: Limitations of Log-Precision Transformers" shows that log-precision transformers lie in log-space-uniform TC^0

**Questions:**

1. The paper claims that showing that transformers are contained in $TC^0$ proves a separation from $P$. However, as far as I know, this is not known? Could you send a reference?

---

### Meta-Review · Area_Chair_uHwb · 2023-12-04

**Metareview:**

This work studies the expressive power of Transformers. They take a circuit-complexity viewpoint, and argue that Transformers cannot express all circuits with only polynomial blowup. Thus, there exist functions which cannot be computed by [a potentially nonuniform family of] Transformers.

Reviewers agree that the problem of formally understanding what Transformers can express and learn is important and timely.
However, all reviewers have observed significant technical issues with this paper, which were unaddressed by authors in rebuttal. Specifically, the main technical results are simple reductions (or re-phrasings) of well-known results in computational complexity. It is unclear what additional insight this paper offers, thus I must recommend rejection. I encourage the authors to better situate their work in the context of known results.

**Justification For Why Not Higher Score:**

All reviewers noted that the technical contribution was minimal, beyond prior work. Reviewer 8H3T summarizes these prior results.

**Justification For Why Not Lower Score:**

N/A

---

### Decision · Program_Chairs · 2024-01-16

Reject